# Look-to-Touch: A Vision-Enhanced Proximity and Tactile Sensor for Distance and Geometry Perception in Robotic Manipulation

Yueshi Dong, Jieji Ren, Zhenle Liu, Zhanxuan Peng, Zihao Yuan, Ningbin Zhang, and Guoying Gu

## I. Novelty/progress claims

Camera-based tactile sensors provide robots with a high-performance tactile sensing approach for environment perception and dexterous manipulation. However, achieving comprehensive environmental perception still requires cooperation with additional sensors, makes the system bulky and limits its adaptability to unstructured environments. In this work, we propose a camera-based dual-model sensor with vision enhancement, which can realize accurate long-distance proximity sensing while simultaneously maintaining ultra-high-resolution texture sensing and reconstruction capabilities. Specifically, in stand of a fixed gel layer, our sensor features a partially transparent sliding window, enabling mechanical switching between tactile and visual modes. For each sensing mode, a proximity sensing model and a contact texture reconstruction model are proposed. Through integration with soft fingers, we systematically evaluate the performance of each mode, also in their synergistic operation. Experimental results show robust distance tracking across various speeds, nanometer-scale roughness detection, and submillimeter 3D texture reconstruction. The combination of both modalities improves the robot's efficiency in executing grasping tasks.

## II. Background/State of the Art

Among various exteroceptive modalities, tactile sensors play a crucial role in enabling robots to perceive contact-related information. Camera-based tactile sensors is one of the most promising approach for environment perception and dexterous operation. [1] [2] [3] [4]. However, most existing designs, being sealed and opaque, are limited to negative-scale perception, restricting external information and, consequently, lacking positive-scale perception capability. To address these problems, some prospective research efforts take an alternative approach to expanding the 'see-through' capability from a single VBTS sensing unit, enabling it to obtain a full-scale sensing ability. However, all of the proposed method sacrifice its effectiveness in detailed surface characterization [5]. The design of the dual-mode sensor that integrates long-distance proximity sensing and high-quality contact sensing has long remained a persistent challenge.

This work was supported by the National Natural Science Foundation of China under Grant 52025057 and Grant 52305029, and in part by the Science and Technology Commission of Shanghai Municipality under Grant 24511103400. *(Corresponding author: Guoying Gu.)*

Yueshi Dong, Jieji Ren, Zhenle Liu, Zhanxuan Peng, Zihao Yuan, Ningbin Zhang and Guoying Gu are with the State Key Laboratory of Mechanical System and Vibration, Shanghai Jiao Tong University, Shanghai, China, and also with the Robotics Institute, School of Mechanical Engineering, Shanghai Jiao Tong University, Shanghai, China (e-mail: dongys525@sjtu.edu.cn; jiejiren@sjtu.edu.cn; extraordinary@sjtu.edu.cn; steven2606@sjtu.edu.cn; yuanzihao@sjtu.edu.cn; zhangnb@sjtu.edu.cn; guguoying@sjtu.edu.cn).

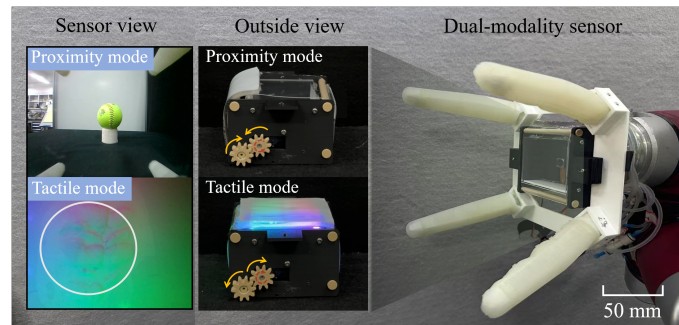

Fig. 1. Illustration of two working modes of the V-T Palm and its application in a robotic system.

## III. Description of the new method or system

In this article, we propose a vision-enhanced proximity and tactile sensing palm with a rotatable belt as the mode-shifting mechanism (see Fig. 1). The belt is partially covered by an opaque elastic layer, serving as the outermost contact interface. In *proximity sensing* mode, this elastomer is removed through controlled rotation, let the inner lens exposed, which then operates purely as an imaging sensor. Through an ingenious structural design, a seamless transition between the two sensing modes is achieved. Under *tactile sensing* mode, the sensor utilizes a photometric stereo vision algorithm [6] to reconstruct the morphology of the contact surface, delineate the contact position, and classify the tactile texture. Under *proximity sensing* mode, sensor carries out real-time distance measurement on the target by applying monocular depth reconstruction on the images of a zoom camera [7], [8], [9], and applies shape segmentation on the target object, which alleviates the complexity of hand-eye calibration and provides essential information for the manipulator's grasp planning.

## IV. Experimental results

The experimental results show an accurate distance measurement across various targets under different speeds in real-time conditions. Furthermore, it can achieve a nanometer-scale roughness resolution and sub-millimeter texture reconstruction. The integration of both modalities improves the robot's efficiency in executing various operation tasks. The system demonstrates strong potential for adaptive grasping. Future work will focus on enhancing the perception framework to support proximity sensing of transparent objects, increasing the upper limit of tactile sensing resolution, and improving adaptability to ultra-soft materials, with an aim to expand the applicability of the proposed sensor in more challenging scenarios.

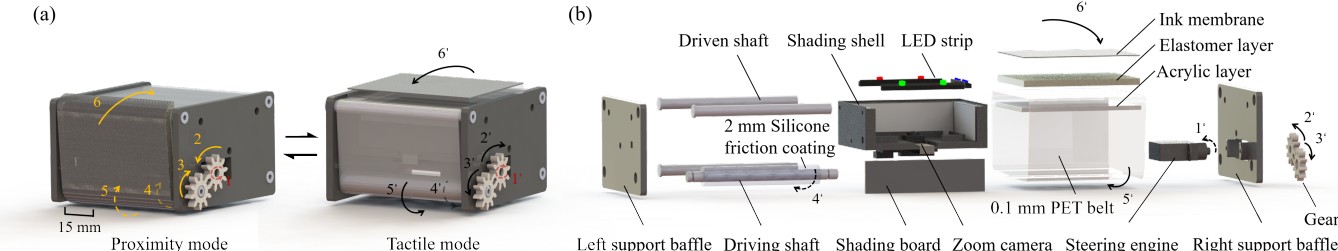

Fig. 2. Structural design of the V-T PALM sensor. (a) Assembly views in proximity and tactile modes, with annotated arrows indicating the internal actuation sequence during bidirectional mode switching. (b) Detailed exploded view illustrating the component layout of the sensor.

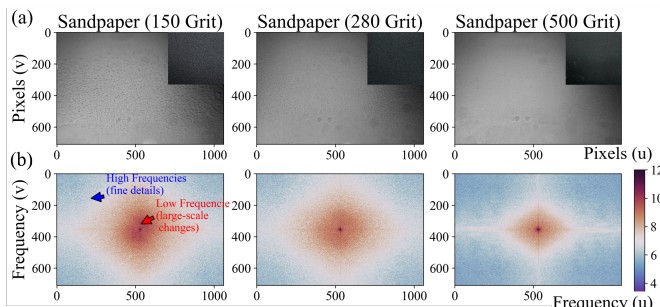

Fig. 3. Frequency domain analysis of tactile images. (a) the grayscale-processed tactile data of sandpapers with different grit sizes, with the real photo of the corresponding sandpaper shown in the upper right corner. (b) the frequency amplitude spectrum visualized on a logarithmic scale.

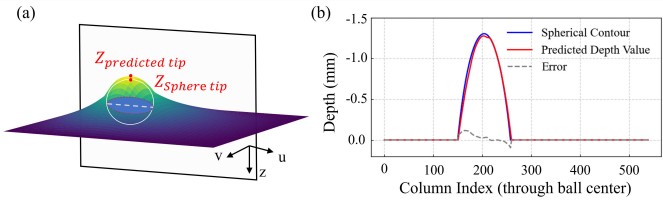

Fig. 4. Characterization of the geometric reconstruction method. (a) Reconstructed surface of the sensor under indentation by a standard 8 mm-diameter sphere. (b) Comparison between the extracted depth profile along the sphere's central column and the theoretical spherical surface.

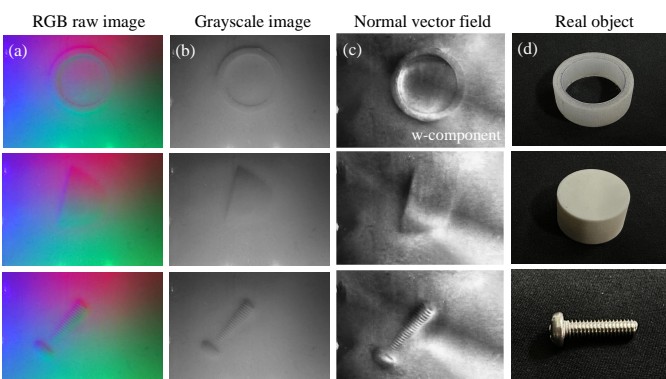

Fig. 5. Reconstruction performance of geometric texture. (a) shows the raw image data captured by the palm sensor. (b) displays the grayscale images of raw data, illustrating geometric information before processing. (c) shows the normal surface in w-axis calculated by the proposed FCNN algorithm, indicates the depth information, and (d) presents the real objects.

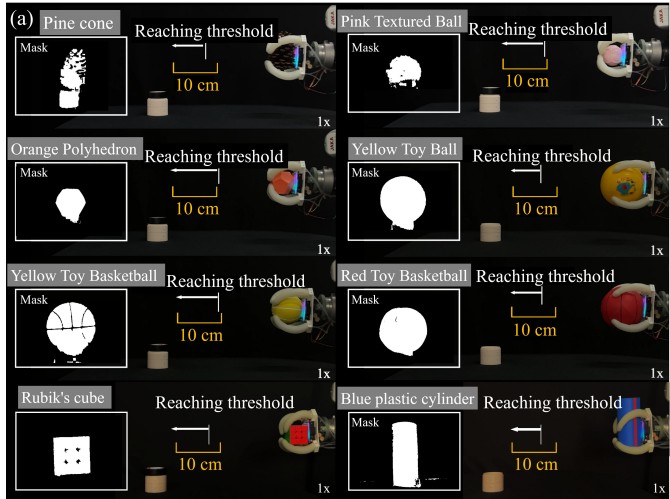

Fig. 6. Experimental results of pre-planned grasping and subtle identification tests, which shows the grasping performance for different objects.

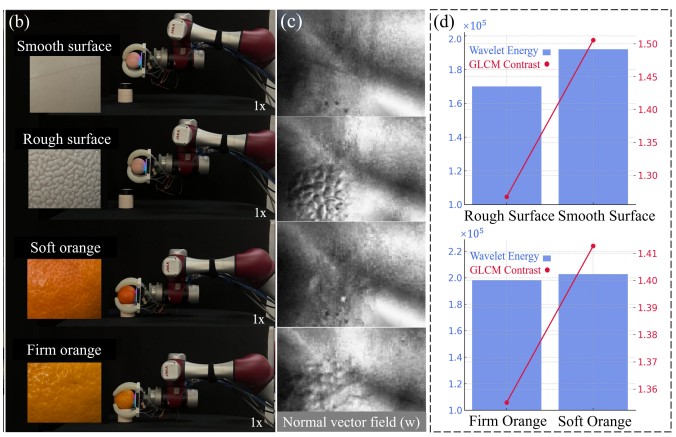

Fig. 7. Experimental results of the reconstruction results and roughness analysis of the tactile palm.

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
