# OpenReview forum: "Look-to-Touch: A Vision-Enhanced Proximity and Tactile Sensor for Distance and Geometry Perception in Robotic Manipulation"
_IEEE.org/IROS/2025/Workshop/Tactile_Sensing — IROS 2025 Workshop Tactile Sensing OralPoster_

### Official Review · Reviewer_Taez · 2025-09-15
**Review on Look-to-Touch**

**Rating:** 7
**Confidence:** 4

**Review:**

The paper presents a novel and interesting sensor design that integrates high-quality contact sensing with proximity sensing, a capability highly relevant to robotic manipulation. The writing is overall clear and the contribution is well-motivated. I do, however, have several suggestions for improvement:

1) Quantitative evaluation of gel rolling/removal

It would strengthen the paper to include quantitative information about the duration required to roll the gel surface on the gel and to remove it. This would give readers a better sense of the sensor’s responsiveness and practical usability.

2) Figure references

The short abstract would benefit from one or two additional sentences referencing the key figures and briefly explaining what they show. This would make the submission more self-contained and help readers quickly grasp the main experimental results or system design at a glance.

3) Expanded related work

The related work section could be enriched by citing more prior efforts that combine proximity and tactile sensing. In particular, the following works are relevant:

Yamaguchi, Akihiko, and Christopher G. Atkeson. "Implementing tactile behaviors using fingervision." 2017 IEEE-RAS 17th International Conference on Humanoid Robotics (Humanoids). IEEE, 2017.

F. R. Hogan, J.-F. Tremblay, B. H. Baghi, M. Jenkin, K. Siddiqi, and
G. Dudek, “Finger-STS: Combined Proximity and Tactile Sensing for
Robotic Manipulation,” IEEE Robotics and Automation Letters, vol. 7,
no. 4, pp. 10 865–10 872, Oct. 2022.

(not directly high spatial resolution, but still multi-modal touch sensing): Mittendorfer, Philipp, and Gordon Cheng. "Humanoid multimodal tactile-sensing modules." IEEE Transactions on robotics 27.3 (2011): 401-410.

---

### Official Review · Reviewer_cLLv · 2025-09-19
**Review of Look-to-Touch paper**

**Rating:** 7
**Confidence:** 4

**Review:**

The paper proposes the design and working principle for an interesting vision-based proximity and tactile sensor. The presentation is comprehensive, showing the key contributions, however, some improvements are desirable.
1. The authors should briefly detail the accuracy of contact detection. For example, what is the error for contact localization, or how to decide if an object is making contact with the sensor?
2. Figure 1 is somewhat confusing. Although it depicts the sensor, the presence of fingers may mislead the reader. The authors should clearly indicate which part corresponds to the sensor.
3. The Background section could include the work "Vi2TaP: A Cross-Polarization Based Mechanism for Perception Transition in Tactile-Proximity Sensing With Applications to Soft Grippers", 10.1109/LRA.2025.3566583.